# GENERATIVE LOW-SHOT NETWORK EXPANSION

## ABSTRACT

Conventional deep learning classifiers are static in the sense that they are trained on a predefined set of classes and learning to classify a novel class typically requires re-training. In this work, we address the problem of *Low-Shot network-expansion* learning. We introduce a learning framework which enables expanding a pre-trained (*base*) deep network to classify novel classes when the number of examples for the novel classes is particularly small. We present a simple yet powerful distillation method where the base network is augmented with additional weights to classify the novel classes, while keeping the weights of the base network unchanged. We term this learning *hard distillation*, since we preserve the response of the network on the old classes to be equal in both the base and the expanded network. We show that since only a small number of weights needs to be trained, the hard distillation excels for low-shot training scenarios. Furthermore, hard distillation avoids detriment to classification performance on the base classes. Finally, we show that low-shot network expansion can be done with a very small memory footprint by using a compact generative model of the base classes training data with only a negligible degradation relative to learning with the full training set.

## 1 INTRODUCTION

In many real life scenarios, a fast and simple *classifier expansion* is required to extend the set of classes that a deep network can classify. For example, consider a cleaning robot trained to recognize a number of objects in a certain environment. If the environment is modified with an additional novel object, it is desired to be able to update the classifier by taking only a few images of that object and expand the robot classifier. In such a scenario, the update should be a simple procedure, based on a small collection of images captured in a non-controlled setting. Furthermore, such a low-shot network update should be fast and without access the *entire training set* of previously learned data. A common solution to classifier expansion is *fine-tuning* the network Kading et al. (2016). However fine-tuning requires keeping a large amount of *base* training data in memory, in addition to collecting sufficient examples of the novel classes. Otherwise, fine-tuning can lead to degradation of the network accuracy on the base classes, also known as *catastrophic forgetting* French (1999). In striking contrast, for some tasks, humans are capable of instantly learning novel categories. Using one or only a few training examples humans are able to learn a novel class, without compromising previously learned abilities or having access to training examples from all previously learned classes.

We consider the classifier expansion problem under the following constraints:

1. Low-shot: very few samples of the novel classes are available.
2. No forgetting: preserving classification performance on the base classes.
3. Small memory footprint: no access to the base classes training data.

In this work we introduce a low-shot network expansion technique, augmenting the capability of an existing (base) network trained on base classes by training additional parameters that enables to classify novel classes. The expansion of the base network with additional parameters is performed in the last layers of the network.

To satisfy low-shot along with no-forgetting constraints, we present a *hard distillation* framework. Distillation in neural networks Hinton et al. (2014) is a process for training a target network to imitate another network. A loss function is added to the target network so that its output matches the output of the mimicked network. In standard *soft distillation* the trained network is allowed to deviate from

the mimicked network. Whereas hard distillation enforces that the output of the trained network for base classes matches the output of the mimicked network as a hard constraint. We achieve hard distillation by keeping the weights of the base network intact, and learn only the newly added weights. Network expansion with hard distillation yields a larger network, distilling the knowledge of the base network in addition to augmented capacity to classify novel classes. We show that in the case of low-shot (only 1–15 examples of a novel class), hard distillation outperforms soft distillation. Moreover, since the number of additional parameters in the expanded network is small, the inference time of the new network is nearly identical to the base network.

To maintain a small memory footprint, we refrain from saving the entire training set. Instead, we present a compact generative model, consisting of a collection of generative models fitted in the feature space to each of the base classes. We use a Gaussian Mixture Model (GMM) with small number of mixtures, and show it inflicts a minimal degradation in classification accuracy. Sampling from the generative GMM model is fast, reducing the low-shot training time and allowing fast expansion of the network.

We define a benchmark for low-shot network expansion. The benchmark is composed of a series of tests of increasing complexity, ranging from simple tasks where base and novel classes are from different domains and to difficult tasks where base and novel classes are from the same domain and shares objective visual similarities. We perform a comprehensive set of experiments on this challenging benchmark, comparing the performance of the proposed to alternative methods.

To summarize, the main contributions of the paper are:

1. A novel hard-distillation solution to a low-shot classifier expansion problem
2. GMM as a sufficient generative model to represent base classes in a feature space
3. A new benchmark for the low-shot classifier expansion problem

## 2 RELATED WORKS

A common solution to the class-incremental learning problem is to use a Nearest-Neighbors (NN) based classifier in feature space. A significant advantage of a NN-based classifier is that it can be easily extended to classify a novel class, even when only a single example of the class is available (*one-shot learning*). However NN-based classifiers require keeping in the memory significant amount of training data from the base classes. Mensink et al. (2013) proposed to use Nearest Class Mean (NCM) classifier, where each class is represented by a single prototype example which is the mean feature vector of all class examples. One major disadvantage of NCM and NN-based methods is that they are based on a fixed feature representation of the data. To overcome this problem Mensink et al. (2013) proposed to learn a new distance function in the feature space using metric learning.

The ideas of metric learning combined with the NN classifier resonate with recent work by Vinyals et al. (2016) on *Matching Networks* for one-shot learning, where both feature representation and the distance function are learned end-to-end with attention and memory augmented networks. The problem we consider in this paper is different from the one discussed by Vinyals et al. (2016). We aim to expand existing deep classifier trained on large dataset to classify novel classes, rather than to create a general mechanism for one-shot learning.

Hariharan & Girshick (2016) presented an innovative low-shot learning mechanism, where they proposed a *Squared Gradient Magnitude* regularization technique for an improved *fixed* feature representation learning designed for low-shot scenarios. They also introduced techniques to hallucinate additional training examples for novel data classes. In contrast, we present a method which aims to maximize performance in low-shot network expansion *given* a fixed representation, allowing *expanding* the representation based on novel low-shot data. Furthermore, in our work, we demonstrate the ability to expand the network without storing the entire base classes training data.

Recently, Rebuffi et al. (2016) proposed iCaRL – (Incremental Classifier and Representation Learning), to solve the class-incremental learning problem. iCaRL is based on Nearest-Mean-of-Exemplars classifier, similar to the NCM classifier of Mensink et al. (2013). In the iCaRL method, the feature representation is updated and the class means are recomputed from a small stored number of representative examples of the base classes. During the feature representation update, the network parameters are updated by minimizing a combined classification and distillation loss. The iCaRL method was

introduced as a class-incremental learning method for large training sets. In Section 4 we discuss its adaptation to low-shot network expansion and compare it to our method.

Rusu et al. (2016) proposed the *Progressive Network* for adding new tasks without affecting the performance of old tasks. They propose *freezing* the parameters that were trained on old tasks and *expand* the network with a additional layers when training a new task. Venkatesan & Er (2016) proposed the *Progressive learning* technique which solves the problem of online sequential learning in extreme learning machines paradigm (OS-ELM). The purpose of their work is to incrementally learn the last fully-connected layer of the network. When a sample from a novel class arrives, the last layer is expanded with additional parameters. The Progressive learning solution updates the last layer only sequentially and only works in the ELM framework (does not update internal layers of the network). In another work Venkatesan et al. (2017) proposed an incremental learning technique which augments the base network with additional parameters in last fully connected layer to classify novel classes. Similar to iCaRL, they perform soft distillation by learning all parameters of the network. Instead of keeping historical training data, they propose *phantom sampling* - hallucinating data from past distribution modeled with Generative Adversarial Networks.

In this work we propose a solution that borrows ideas from freeze-and-expand paradigm, improved feature representation learning, network distillation and modeling past data with a generative model. We propose to apply expansion to the last fully connected layer of a base network to enable classification on novel classes, and to deeper layers to extend and improve the feature representation. However, in contrast to other methods Rebuffi et al. (2016); Venkatesan & Er (2016), we do not retrain the base network parameters, but only the newly introduced weights of the expansion.

Moreover, the extended feature representation is learned from samples of base and novel classes. In contrast to Hariharan & Girshick (2016), where the improved feature representation is learned from simulating low-shot scenarios on the base classes only, before the actual novel data is available. Finally, in order to avoid keeping all historical training data, we use Gaussian Mixture Model of the feature space as a generative model for base classes.

## 3 THE PROPOSED METHOD

Assume a deep neural network is trained on $K$ base classes with the full set of training data. This base network can be partitioned into two subnetworks: a feature extraction network and a classification network. The feature extraction network $f_{rep}$ maps an input sample $x$ into a feature representation $v \in \mathbb{R}^N$. The classification network $f_{cls}$ maps feature vectors $v$ into a vector of approximated class posterior probabilities $P(k|v)$ which correspond to each one of $K$ classes. The whole network can be represented as composition of two networks $f_{net}(x) = f_{cls}(f_{rep}(x))$. For example, if the classification network consists of the last fully connected layer (FC) followed by softmax, then $f_{cls}(v)[k] = \frac{1}{Z}e^{w_k^T v}$, where $w_k \in \mathbb{R}^N$ is class's $k$ weights vector, and $Z$ is the normalization factor.

In the following we discuss how the pre-learned feature representation of feature extraction network can be leveraged to classify additional classes in low-shot scenario with only relatively minor changes to the classification subnetwork.

### 3.1 EXPANSION OF THE LAST LAYER OF CLASSIFICATION SUBNETWORK

First, we discuss how to expand the classification network to classify one additional class. We can expand $f_{cls}$ from a $K$-class classifier into $K + 1$ class classifier by adding a new weight vector $w_{K+1} \in \mathbb{R}^N$ to the last FC layer. Thus, the $K + 1$ class probability is $f_{cls}(v)[K+1] = \frac{1}{Z'}e^{w_{K+1}^T v}$, where $Z'$ is a new normalization factor for $K + 1$ classes. We would like to preserve classification accuracy on the base classes to avoid catastrophic forgetting. To that end, during training we constrain to optimize of the $w_{K+1}$ weights, while the vectors $\{w_i\}_{i=1}^K$ are kept intact. We refer to this paradigm as *hard distillation*. By preserving the base classes weight vectors, we guarantee that as a result of the last classification layer expansion the only new errors that can appear are between the novel class and the base classes, but not among the base classes. Moreover, the small number of newly learned parameters helps avoid over-fitting, which is especially important in low-shot scenarios.

Similarly, we can expand the classification network to classify more than one novel class.

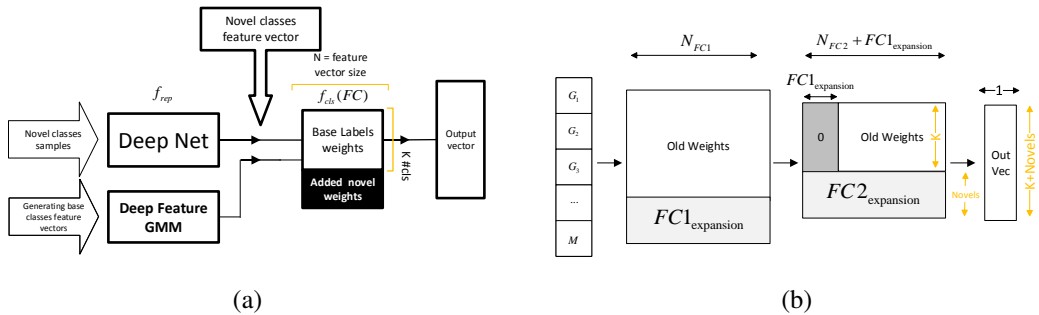

(a)                                                      (b)

Figure 1: (a) Gen-LSNE overview, generating $f_{rep}$ feature representation of base classes to train the $f_{cls}$ expansion. (b) training last two layers, learning shared representation in addition to the per novel class weights expansion: $G_i$ are samples of feature vector generations of base class $i$, $M$ are the novel class feature vector measurements, $N_{FC1}$ are the number of input features to $FC1$, $N_{FC2}$ are the number of input feature to $FC2$ before the expansion.

## 3.2 DEEP FEATURE GMM - GENERATIVE MODEL FOR BASE CLASSES

Due to the small memory footprint constraint, we are unable to keep the entire training data of the base classes. As an alternative, we can use a generative model of the base classes and during training draw samples from the model. There are various approaches to this task, such as GAN Goodfellow et al. (2014), VAE Pu et al. (2016), Pixel CNN van den Oord et al. (2016), or conventional methods of non-parametric kernel density estimation Hastie et al. (2001). However, it is usually hard to generate *accurate* samples from past learned distributions in the *image* domain, and these methods still require a significant amount of memory to store the model network parameters. Furthermore, since training typically requires thousands of samples, we prefer a generative model that allows fast sampling to reduce the low-shot phase training time.

In our work, we use the Gaussian Mixture Model (GMM) density estimator as an approximate generative model of the data from the base classes. However, instead of approximating the generative distribution of the *image* data, we approximate a class conditional distribution of its feature representation. Thus, we model a GMM $P(v|c=k) = \sum_{i=1}^{M} \pi_i \mathcal{N}(v|\mu_i, \Sigma_i)$, where $M$ is the number of mixtures for each bass class. In order to satisfy the small memory footprint constraint, we use a GMM which assumes feature independence, *i.e.,* the covariance matrix $\Sigma_i$ of each Gaussian mixture is diagonal. We denote this model as *Deep Feature GMM*. If we have $K$ classes, and the feature vectors dimensionality is $N$, the memory requirements for storing information about base classes is $O(MKN)$. The feature representation $v$, which we learn a generative model for, can be from the last fully connected layer or from deeper layers. In Section 4.5, we evaluate the effectiveness of the use of the Deep Features GMM, showing that despite its compact representation, there is a minimal degradation in accuracy when training a classifier based only on data that is generated from the Deep Features GMM, compared to the accuracy obtained on the full training data.

## 3.3 LOW-SHOT TRAINING

We apply standard data augmentation (random crop, horizontal flip, and color noise) to the input samples of the novel classes and create 100 additional samples variants from each of the novel class samples. These samples are passed through the feature extraction network $f_{rep}$ to obtain their corresponding feature representation. Note that new samples and their augmented variants are passed through $f_{rep}$ only once.

As described in Section 3.1, we expand the classification subnetwork $f_{cls}$ and train the expanded network to classify novel classes in addition to the base classes. Figure 1(a) illustrates the proposed method in the case where $f_{cls}$ is the last fully connected layer. As mentioned above, we only learn the $N$ dimensional vector $w_{K+1}$, which augments the $K \times N$ weight matrix of the FC layer.

Each training batch is composed of base classes feature vectors drawn from the Deep Features GMM models learned from the base classes training data and the available samples of a novel class. The training batch is balanced to have equal number of generations/samples per class.

Since the forward and backward passes are carried out by only the last FC layers, each iteration can be done very rapidly. We use SGD with *gradient dropout* (see below) to learn $w_{K+1}$. More specifically, the weights update is done by:

$$g_{K+1} = g_{K+1} + \mu \Delta w_{K+1}$$
$$w_{K+1} = w_{K+1} - \alpha M g_{K+1}$$

where $\mu$ is the momentum factor, $\alpha$ is the learning rate and $M$ is a binary random mask with probability $p$ of being 1 ($M$ is randomly generated throughout the low-shot training). That is, the gradient update is applied to a random subset of the learned weights. This SGD with gradient dropout is our heuristic variant for Noisy SGD proposed by Ge et al. (2015) which provably helps to escape saddle points. In Section 4.3 we demonstrate the contribution of the gradient dropout when only a few novel labeled samples are available.

### 3.4 Expansion of Deeper Layers for Learning Representation

The procedure described in the previous subsections expands the last classification layer, but does not change the feature representation space. In some cases, especially in those which the novel classes are similar to the base classes, it is desirable to update and *expand* the feature representation.

To *expand* the feature representation, we add new parameters to deeper layers of the network. This of course requires an appropriate expansion of all subsequent layers. To satisfy the hard distillation constraints, we enforce that the feature representation expansion does not affect the network output for the base classes. All weights in subsequent layers which connects the expanded representation to the base classes are set to zero and remain unchanged during learning. In Figure 1(b) we demonstrate an expansion of two last fully connected layers. The $FC2$ weight matrix is zero padded to adjust to the new added weights in $FC1$. Only the expansion to $FC2$ uses the new added features in $FC1$. The details of the representation learning expansion can be found in Appendix D.

## 4 Experiments

In this section, we evaluate the proposed low-shot network expansion method on several classification tasks. We design a benchmark which measures the performance of several alternative low-shot methods in scenarios that resemble real-life problems, starting with easier tasks (Scenario 1) to harder tasks (Scenario 2 & 3). In each experiment, we use a standard dataset that is partitioned to base classes and novel classes. We define three scenarios:

Scenario 1 **Generic novel classes:** unconstrained novel and base classes which can be from different domains.

Scenario 2 **Domain specific with similar novel classes:** base and novel classes are drawn from the same domain and the novel classes share visual similarities among themselves.

Scenario 3 **Domain specific with similar base and novel classes:** base and novel classes are drawn from the same domain and each novel class shares visual similarities with one of the base classes.

In each scenario we define five base classes (learned using the full train set) and up to five novel classes, which should be learned from up to 15 samples only. We compare the proposed method to several alternative methods for low-shot learning described in Section 4.2.

### 4.1 Datasets for Low-Shot Network Expansion scenarios

**Dataset for Scenario 1** For the task of generic classification of the novel classes we use the ImageNet dataset Russakovsky et al. (2015), such that the selected classes were not part of the ILSVRC2012 1000 classes challenge. Each class have at least 1000 training images and 250 test images per class. The randomly selected 5 partition of 5 base classes and 5 novel classes.

**Dataset for Scenario 2 and Scenario 3** For these scenarios we use the UT-Zappos50K Yu & Grauman (2014) shoes dataset for fine-grained classification. We choose 10 classes representing different types of shoes each having more than 1,000 training images and 250 test images.

To define similarity between the chosen classes, we fine-tune the base network (VGG-19 Simonyan & Zisserman (2015)) on the selected classes with the full dataset, and we use the confusion matrix as a measure of similarity between classes. Using the defined similarities, we randomly partition the 10 classes to 5 base and 2 novel classes, where for Scenario 2 we enforce similarity between novel classes, and for Scenario 3 we enforce similarity between novel and base classes. The confusion matrix representing the visual similarities and an illustration of the similarities between the base and novel classes is presented in Figure 10 in Appendix C.

## 4.2 EVALUATED METHODS

In the proposed method we use the VGG-19 network Simonyan & Zisserman (2015) trained on ImageNet ILSVRC2012 Russakovsky et al. (2015) 1000 classes as a feature extraction subnetwork $f_{rep}$. In all three scenarios for training the classification subnetwork $f_{cls}$ on the base classes, we fine-tune the last two fully-connected layers of VGG-19 on the 5 selected base classes, while freezing the rest of the layers of $f_{rep}$.

We denote the method proposed in Section 3 as Generative Low-Shot Network Expansion: *Gen-LSNE*. We compare our proposed method to NCM Mensink et al. (2013), and to the *Prototype-kNN* method which is an extension of NCM and the soft distillation based method inspired by iCaRL method Rebuffi et al. (2016), adapted for the low-shot scenario.

### 4.2.1 NCM & PROTOTYPE-KNN

We compare the proposed method to NCM classifier proposed by Mensink et al. (2013). Additionally, we extend the NCM classifier by using multiple prototypes for each class, as in the *Prototype-kNN* classifier Hastie et al. (2001). Both NCM and Prototype-kNN are implemented in a fixed feature space of the FC2 layer of the VGG-19 network. In our implementation of the Prototype-kNN, we fit a Deep Features GMM model with 20 mixtures for each of the base classes. We extract feature representation of all of the available samples from the novel classes. The Deep Features GMM centroids of the base feature vectors and the novel feature vectors of the samples are considered as a prototypes of each class. We set $k$ for Prototype-kNN classifier to be the smallest number of prototypes per class (the number of prototypes in the novel classes is lower than the number of mixture in the base classes). The Prototype-kNN classification rule is the majority vote among $k$ nearest neighbors of the query sample. If the majority vote is indecisive, that is, there are two or more classes with the same number of prototypes among the $k$ nearest neighbors of query image, we repeat classification with $k = 1$.

### 4.2.2 LOW-SHOT WITH SOFT DISTILLATION

We want to measure the benefit of the *hard distillation* constraint in low-shot learning scenario. Thus, we formulate a soft distillation based method inspired by iCaRL Rebuffi et al. (2016) and methods described by Venkatesan et al. (2017) and Venkatesan & Er (2016) as an alternative to the proposed method.

In the iCaRL method, feature representation is updated by re-training the whole representation network. Since in low-shot scenario we have only a small number of novel class samples, updating the whole representation network is infeasible. Using the soft distillation method, we adapt to the low-shot scenario by updating only the last two fully connected layers $FC1, FC2$, but still use a combination of distillation and classification loss as in the iCaRL method.

The iCaRL method stores a set of prototype images and uses the Nearest Mean Exemplar (NME) classifier at the final classification stage. In order to provide a fair comparison with the hard distillation method and uphold our memory restriction, we avoid storing prototypes in image domain, and use the proposed Deep-Features GMM as a generative model for the base-classes. Using NME classifier with prototypes of the base classes is in fact a special case of Prototype-kNN with $k = 1$. Therefore, in soft distillation method instead of NME we use a learned expanded network with additional parameters in last fully connected layers, which aligns with Venkatesan et al. (2017) and Venkatesan & Er (2016), and in our proposed hard-distillation method in Section 3.1.

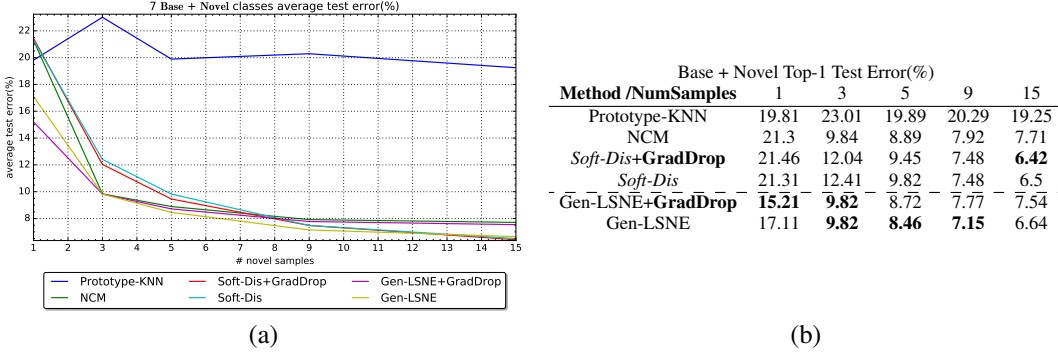

Figure 2: **Scenario 1, Generic novel classes**: Top-1 Test Error on the proposed-method, Prototype-kNN, NCM and *Soft-Dis*: (a) the average Test Error on all 7 classes (base + novel), (b) show the results viewed in (a) in tabular form.

To summarize, soft distillation applies a distillation loss and allows the $FC1, FC2$ layers to adjust to the new data, while the proposed hard-distillation freezes $FC1, FC2$ and trains only the new (expanded) parameters without using a distillation loss. We denote the soft distillation based methods as *Soft-Dis* in the presented results.

### 4.2.3 GRADIENT DROPOUT

In Section 3.3 we proposed using gradient dropout regularization on SGD as a technique to improve convergence and overcome overfitting in a low-shot scenario. We perform ablation experiments to assess the importance of the gradient dropout and train using both soft distillation (Soft-Dis) and proposed hard distillation (Gen-LSNE) with and without gradient dropout regularization.

### 4.3 RESULTS: EXPANSION OF THE LAST FULLY CONNECTED LAYER

**Scenario 1: Generic novel classes**   In this experiment, the base classification network is trained on five base classes and then expanded to classify two novel classes chosen at random. For each of the five class partitions (Section 4.1), we perform five trials by randomly drawing two novel classes from five novel classes available in the partition. The results are an average of 25 trials. The results of this experiment are presented in Figure 2. In Figure 6 in Appendix B we present detailed results of the test error on the base and novels classes apart. Prototype-kNN and the *Soft-Dis* methods perform better on the base classes. However, our method is significantly better on the novel classes and the overall test error is considerably improved, particularly when the number of samples is small. In addition, we see the significant gain in accuracy delivered by the gradient dropout when the number of novel samples is lower than 3 samples. Furthermore, gradient dropout also improves the results of the *Soft-Dis* method.

The Prototype-kNN method is unable to effectively utilize the new available novel samples, however it best preserves the base class accuracy when the number of novel samples and base class prototypes is high (above 5 , see Figure 6). Since the number of prototypes used is equal to the number of novel samples used, the addition of a novel samples/base class prototypes generally has a greater impact on the preservation of the base class accuracy. We assume that since the spread of the base class prototypes is better than that of the novel classes, then some novel samples are misclassified as some similar base class. NCM generally performs considerably better than Prototype-kNN, despite the use of less information from the base classes. However, NCM is unable to effectively utilize more novel samples when they are available. Gen-LSNE significantly outperforms NCM with a single novel sample, and overall outperforms all the tested method with nine and below samples per novel class.

**Scenario 2 & 3: Domain specific with similar novel-to-novel and novel-to-base classes**   As described in Section 4.1, in each scenario we have 5 partitions with five base classes and two novel classes. The results are an average of 5 trials. The result of the experiments are presented in Figures 3 and 4. In Scenario-2 & Scenario-3 we see that the proposed method consistently outperforms the

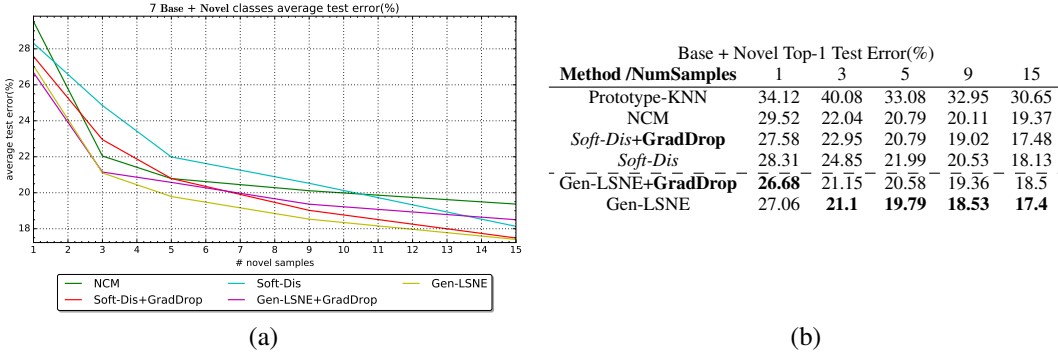

(a)

Base + Novel Top-1 Test Error(%)

| Method /NumSamples | 1 | 3 | 5 | 9 | 15 |
|---|---|---|---|---|---|
| Prototype-KNN | 34.12 | 40.08 | 33.08 | 32.95 | 30.65 |
| NCM | 29.52 | 22.04 | 20.79 | 20.11 | 19.37 |
| *Soft-Dis*+**GradDrop** | 27.58 | 22.95 | 20.79 | 19.02 | 17.48 |
| *Soft-Dis* | 28.31 | 24.85 | 21.99 | 20.53 | 18.13 |
| Gen-LSNE+**GradDrop** | **26.68** | 21.15 | 20.58 | 19.36 | 18.5 |
| Gen-LSNE | 27.06 | **21.1** | **19.79** | **18.53** | **17.4** |

(b)

Figure 3: **Scenario 2, Domain specific with similar novel classes**: Top-1 test error rate on the proposed-method, Prototype-kNN, NCM and *Soft-Dis*: (a) the average test error on all 7 classes (base + novel), (b) show the results viewed in (a) in tabular form and adding *Prototype-kNN*.

*Soft-Dis*, NCM and Prototype-kNN methods. Training Gen-LSNE with gradient dropout improves results in cases with 1 & 3 novel samples per class, especially in Scenario-3. Additionally, training with gradient dropout improves the results of the *Soft-Dis* method. In Figures 7 and 8 in Appendix B we present detailed results of the test error on base and novels classes apart.

## 4.4 RESULTS: EXPANSION OF DEEPER LAYERS FOR LEARNING REPRESENTATION

In this section we explore the effect of expansion of deeper layers, as described in Section 3.4. We partition the datasets as defined in 4.1 to five base and five novel classes, and we test a 10 classes classification task. We expand the feature representation which is obtained after $FC1$ layer with 5 new features. The size of the feature representation after the FC1 layer of VGG-19 is of dimension 4k. Thus, $FC1$ is expanded with $4k \cdot 5$ new weights. The results are averaged over the 5 trails. Figure 5 shows the results obtained, we denote *+5Inner* as the experiments with the additional five shared representation features.

We see a marginal gain in Scenario 1. However, we observe a significant gain in Scenario 2 and 3 when the number of samples increases (especially Scenario 2). Observe that Gen-LSNE significantly outperforms the alternative tested method in almost all of the tested cases. Note that the addition of the five shared additional representation features has no observable effect on *Soft-Dis*.

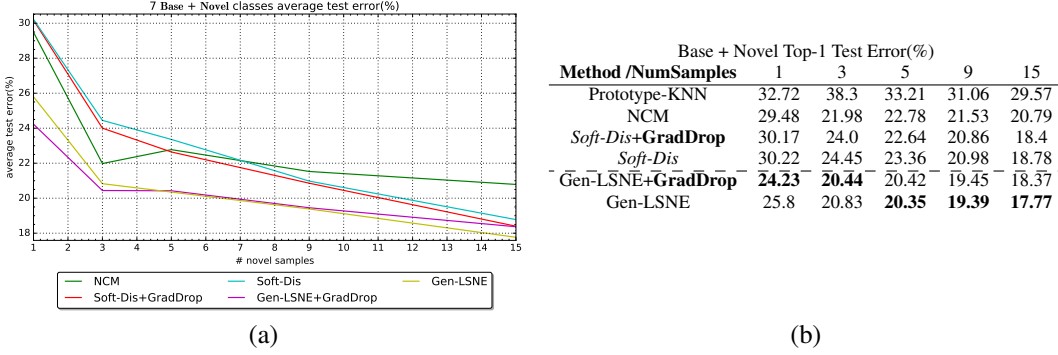

(a)

Base + Novel Top-1 Test Error(%)

| Method /NumSamples | 1 | 3 | 5 | 9 | 15 |
|---|---|---|---|---|---|
| Prototype-KNN | 32.72 | 38.3 | 33.21 | 31.06 | 29.57 |
| NCM | 29.48 | 21.98 | 22.78 | 21.53 | 20.79 |
| *Soft-Dis*+**GradDrop** | 30.17 | 24.0 | 22.64 | 20.86 | 18.4 |
| *Soft-Dis* | 30.22 | 24.45 | 23.36 | 20.98 | 18.78 |
| Gen-LSNE+**GradDrop** | **24.23** | **20.44** | 20.42 | 19.45 | 18.37 |
| Gen-LSNE | 25.8 | 20.83 | **20.35** | **19.39** | **17.77** |

(b)

Figure 4: **Scenario 3, Domain specific with similar class in base**: Top-1 test error rate on the proposed-method, Prototype-kNN, NCM and *Soft-Dis*, showing the gain obtained by applying gradient dropout on the proposed method and on *Soft-Dis*: (a) the average test error on all 7 classes (base + novel). (b) tabular form and adding *Prototype-kNN*

## Scenario 1: Generic novel classes

| | Base + Novel Top-1 Test Error(%) | | | | | | | |
|---|---|---|---|---|---|---|---|---|
| **Method /#Samples** | 1 | 2 | 3 | 5 | 7 | 9 | 11 | 15 |
| Prototype-KNN | 37.76 | 38.49 | 36.98 | 36.66 | 35.36 | 33.97 | 33.45 | 33.23 |
| NCM | 36.08 | 22.79 | 17.18 | 15.54 | 14.96 | 13.7 | 13.37 | 13.17 |
| *Soft-Dis* | 37.39 | 26.34 | 20.46 | 15.6 | 14.04 | 12.18 | 11.4 | 10.62 |
| *Soft-Dis*+5Inner | 37.71 | 26.32 | 20.83 | 15.69 | 14.09 | 12.16 | 11.32 | 10.59 |
| Gen-LSNE | **28.51** | **20.92** | 17.15 | 14.55 | 13.35 | 11.98 | 11.27 | 10.9 |
| Gen-LSNE+5Inner | 28.8 | 21.0 | **17.14** | **14.46** | **13.15** | **11.7** | **11.13** | **10.44** |

(a)

## Scenario 2: Domain specific with similar novel classes

| | Base + Novel Top-1 Test Error(%) | | | | | | | |
|---|---|---|---|---|---|---|---|---|
| **Method /#Samples** | 1 | 2 | 3 | 5 | 7 | 9 | 11 | 15 |
| Prototype-KNN | 41.28 | 40.37 | 39.63 | 39.96 | 39.11 | 37.31 | 38.06 | 36.42 |
| NCM | 41.61 | 36.6 | **32.74** | 29.16 | 28.02 | 27.49 | 27.57 | 27.24 |
| *Soft-Dis* | 41.0 | 38.49 | 36.75 | 31.26 | 28.96 | 27.58 | 27.01 | 25.78 |
| *Soft-Dis*+5Inner | 41.3 | 38.43 | 37.18 | 31.53 | 28.98 | 27.39 | 27.06 | 25.57 |
| Gen-LSNE | **38.52** | **35.95** | 33.2 | 29.09 | 27.47 | 26.42 | 26.71 | 25.87 |
| Gen-LSNE+5Inner | 39.11 | 36.45 | 33.95 | **28.89** | **26.74** | **25.62** | **25.76** | **24.99** |

(b)

## Scenario 3: Domain specific with similar class in base

| | Base + Novel Top-1 Test Error(%) | | | | | | | |
|---|---|---|---|---|---|---|---|---|
| **Method /#Samples** | 1 | 2 | 3 | 5 | 7 | 9 | 11 | 15 |
| Prototype-KNN | 48.69 | 46.81 | 48.7 | 47.96 | 45.02 | 43.72 | 44.17 | 44.35 |
| NCM | 48.91 | 38.62 | 34.47 | 31.98 | 30.72 | 29.91 | 29.34 | 29.49 |
| *Soft-Dis* | 49.47 | 42.8 | 39.49 | 35.18 | 32.64 | 29.97 | 29.39 | 27.75 |
| *Soft-Dis*+5Inner | 49.52 | 42.64 | 39.45 | 35.35 | 32.42 | 30.22 | 29.36 | 27.81 |
| Gen-LSNE | **41.0** | **34.29** | **32.5** | **29.93** | 28.43 | 27.26 | 26.52 | 25.93 |
| Gen-LSNE+5Inner | 42.07 | 34.37 | 33.45 | 30.07 | **28.09** | **26.58** | **26.12** | **25.7** |

(c)

Figure 5: **Expansion of Deeper Layers for Learning Representation:** showing performance obtained with learning additional 5 shared inner features, *+5Inner* marks the addition of the shared expanded features: (a) averaged results on Scenario 1 , (b) averaged results on Scenario 2, (c) averaged results on Scenario 3.

### 4.5 RESULTS: DEEP-FEATURES GMM EVALUATION

In the Deep-features GMM evaluation experiment, we feed the full training data to the base network and collect the feature vectors before $FC1$, *i.e.,* two FC layers before the classification output. We fit a GMM model to the feature vectors of each of the base classes with a varying number of mixture. We train the two last FC layers of the base network from randomly initialized weights, where the training is based on generating feature vectors from the fitted GMM. We measure the top-1 accuracy on the test set of the networks trained with GMM models and the base network trained with full training data

| Dataset /# Mixtures | | Top-1 Accuracy(%) | | | | |
|---|---|---|---|---|---|---|
| | Full | 1 | 10 | 20 | 40 | 60 |
| imagenet-group1-base | 95.3 | 91.94 | 94.03 | 94.19 | 94.03 | **94.57** |
| imagenet-group2-base | 98.0 | 93.83 | 97.04 | 96.63 | 96.54 | **97.37** |
| imagenet-group3-base | 98.2 | 94.40 | 96.81 | **97.45** | 97.09 | 96.52 |
| imagenet-group4-base | 98.8 | 95.60 | 98.16 | 98.01 | 98.30 | **98.58** |
| imagenet-group5-base | 99.0 | 97.26 | 98.26 | 98.01 | 98.01 | **98.26** |
| ut-zap-scenario3-base | 89.5 | 73.23 | 85.34 | 85.10 | **85.50** | **85.50** |
| ut-zap-scenario2-novel | 86.5 | **81.81** | 80.59 | 78.97 | 78.92 | 81.27 |
| ut-zap-scenario2-base | 91.9 | 82.15 | 87.73 | 88.45 | **91.16** | 90.68 |

Table 1: Deep-Features GMM Evaluation: Full stands for Fine tuning FC7,FC8 with the full training data.

on the datasets defined in 4.1. The difference in top-1 accuracy between the network trained with full data and the networks trained with GMM models represent degradation caused by compressing the data into simple generative model. The results of the experiment presented in the Table 1 demonstrate that learning with samples from GMM models cause only a negligible degradation relative to learning with a full training set. We observe that the degradation in accuracy is not monotonic in the number of GMM mixtures, and that for many practical purposes a small number of mixture may be sufficient.

## 5 CONCLUDING REMARKS

We have introduced Gen-LSNE , a technique for low-shot network expansion. The method is based on hard-distillation, where pre-trained base parameters are kept intact, and only a small number of parameters are trained to accommodate the novel classes. We presented and evaluated the advantages of hard-distillation: (i) it gains significant increased accuracy (up to $20\%$) on the novel classes, (ii) it minimizes forgetting: less than $3\%$ drop in accuracy on the base classes, (iii) small number of trained parameters avoids overfitting, and (iv) the training for the expansion is fast. We have demonstrated that our method excels when only a few novel images are provided, rendering our method practical and efficient for a quick deployment of the network expansion.

We have also presented Deep–Features GMM for effective base class memorization. This computationally and memory efficient method allows training the network from a generative compact feature-space representation of the base classes, without storing the entire training set.

In basic setting of a hard distillation, novel class classification is based on the base representation. We have also presented an extended version of network expansion, where the representation is updated by training the last deeper layers. We have shown that when more images of the novel classes are available, the adjusted representation is effective. Generally speaking, the representation can be improved as more samples are available, but at the expense of longer training, larger memory footprint, and risking forgetting.

In the future we would like to continue exploring hard-distillation methods, in extremely low-shot classifier expansion, where only one or a handful of images of the novel class are provided. An additional research direction would be to maximize the information gain that is obtained from each novel image, aspiring towards human low-shot understanding.

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

# APPENDIX A    VGG-19 BASE NETWORK TRAINING HYPER PARAMETERS

We use batch size of 60, and 5000 training iterations. We apply SGD with momentum $\mu = 0.9$ and with learning rate $lr = 0.001$. We apply polynomial learning rate policy with $power = 0.25$, and $L2$ norm weight decay of $0.0005$. The network was fine-tuned on two TITAN-X Pascal GPUs.

# APPENDIX B    BASE & NOVEL CLASSES TEST ERROR

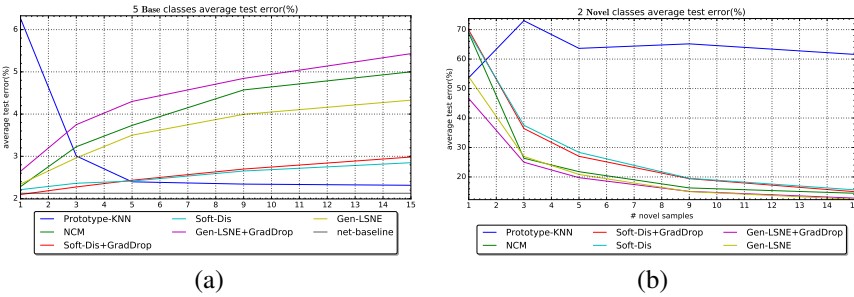

Figure 6: **Scenario 1, Generic novel classes**: Top-1 test error rate on the proposed-method, Prototype-kNN, NCM and *Soft-Dis*: (a) the test error accuracy on the 5 base classes in a 7 class classification task, (b) the average test error on the 2 novel classes in a 7 class classification task

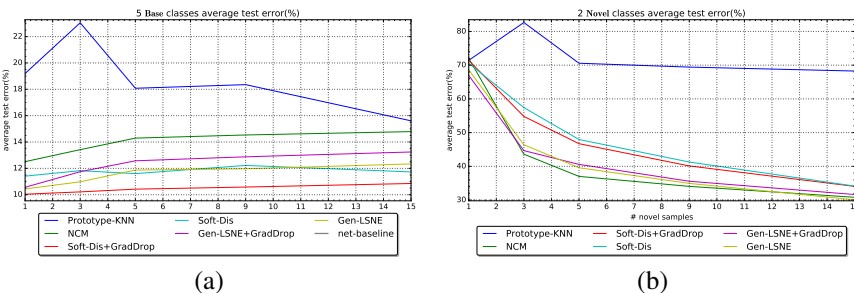

Figure 7: **Scenario 2, Domain specific with similar novel classes**: Top-1 test error rate on the proposed-method, Prototype-kNN, NCM and *Soft-Dis*: (a) the average test error on the 5 base classes in a 7 class classification task, (b) the average test error on the 2 novel classes in a 7 class classification task.

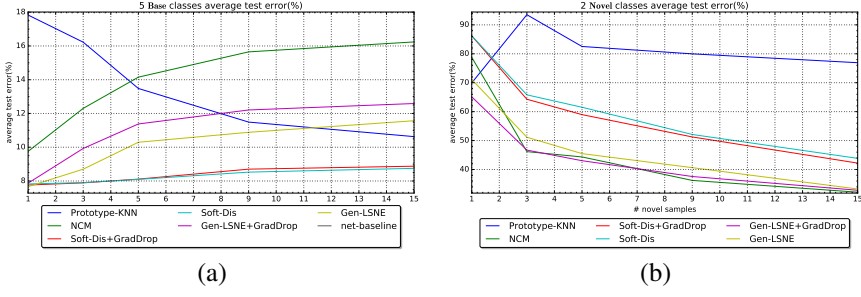

Figure 8: **Scenario 3, Domain specific with similar class in base**: Top-1 test error rate on the proposed-method, Prototype-kNN, NCM and *Soft-Dis* : (a) the average test error on the 5 base classes in a 7 class classification task, (b) the average test error on the 2 novel classes in a 7 class classification task.

## APPENDIX C    CLASSES AND PARTITIONS OF UT-ZAPPOS50K YU & GRAUMAN (2014) DATASET

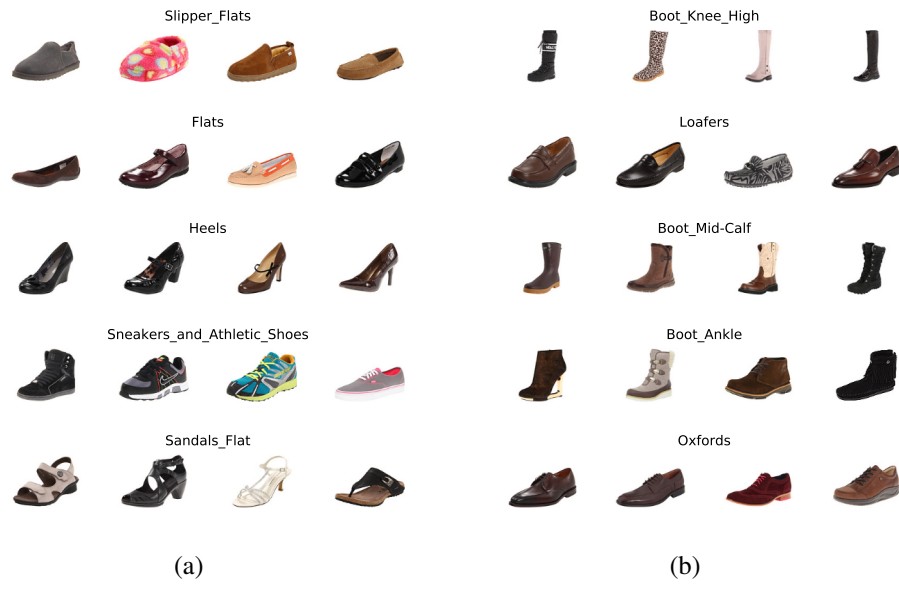

(a)                                                        (b)

Figure 9: **Scenario 2**, Partition with similar novel classes: (a) random examples from base classes; (b) random examples from novel classes. For example, in Scenario 2 we aim to distinguish between Loafers and Oxfords based on Low-Shot samples, with the base classes shown in (a)

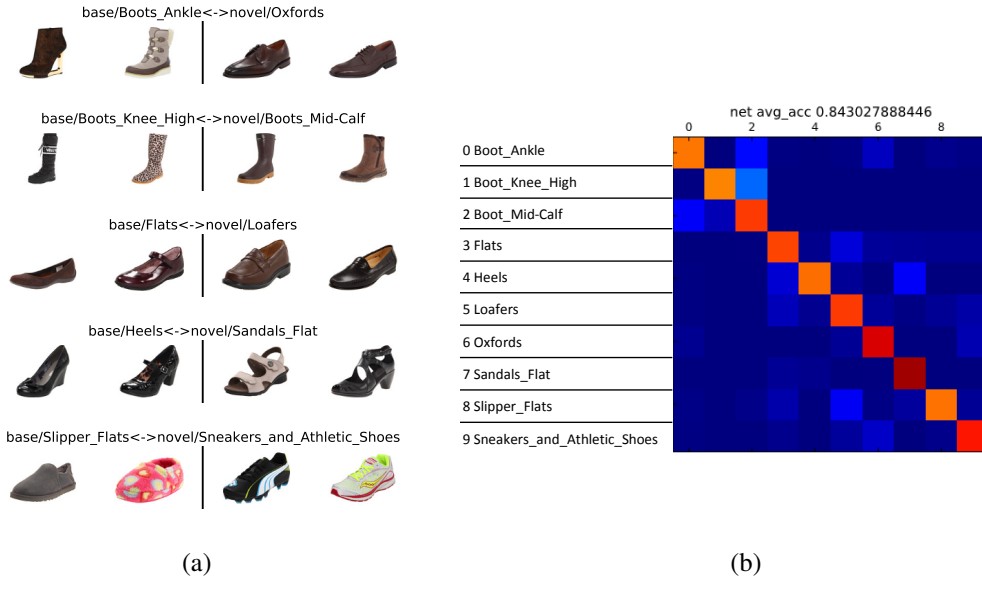

(a)                                                        (b)

Figure 10: (a) Partition with similarities between base to novel classes. (b) Confusion matrix of 10 UT-Zappos50K classes based on fine-tuned VGG-19 Simonyan & Zisserman (2015) network.

## APPENDIX D    REPRESENTATION LEARNING: WEIGHTS INITIALIZATION

The fully connected layer FC1 is parametrized with weight matrix $W \in \mathbb{R}^{N \times V}$, where $V$ is the dimensionality of input feature vector $\nu$ and $N$ is the dimensionality of the output feature representation vector $v$. As was described in Section 3.4 we want to extend the feature representation $v \in \mathbb{R}^N$ with $E$ additional dimensions $\tilde{v} \in \mathbb{R}^E$. We expand the weight matrix $W$ with $E \times V$ additional weights as shown in Figure 1(b). We denote the expanded weights as $W_{exp} \in \mathbb{R}^{E \times V}$. The expanded weights are to be learned from the novel data.

We draw a random set of $S$ novel examples. Let

$$R = \frac{1}{S} \sum_{i=1}^{S} W \cdot \nu_i$$

denote the mean response of the FC1 to the set of novel examples. Let $\{j_i\}_{i=1}^{E}$ be the $E$ indexes of maximum elements of $R$. We initialize the expansion to FC1 layer $W_{exp}$ in the following manner:

$$W_{exp} = \alpha [w_{j_1}^T, w_{j_2}^T, ...w_{j_E}^T]^T + (1 - \alpha)\epsilon$$

where $w_j \in \mathbb{R}^V$ is the $j'th$ row of matrix $W$, $\epsilon \sim \mathcal{N}(0, std(W))$, and $\alpha$ is a weight constant (in our experiments $\alpha = 0.25$). This initialization allows the expansion of the feature representation $\tilde{v}$ to have non zero responses (after ReLU) with respect to the novel samples. Since we operate in a Low-Shot scenario, where only few samples of novel classes are available, this weights initialization plays crucial role in convergence of FC1 extended weights.

We initialize the subsequent layer FC2 in the following manner: let $W' \in \mathbb{R}^{K \times N}$ be the weight matrix of FC2, where $N$ is the dimensionality of the feature representation vector $v$ and $K$ is the number of base classes. Since $v$ was expanded with additional $E$ features $\tilde{v}$, and we want to allow classification of $L$ novel classes, the dimension of expanded $W'$ will be $(K + L) \times (N + E)$. As was mentioned in Section 3.4 and illustrated in Figure 1(b), the hard distillation constraint requires that $W'$ will be zero-padded with $K \times E$ zeros to avoid influence of the expanded features $\tilde{v}$ on the output of the base network. In contrast, the expansion of $W'$ which we denote $W'_{exp} \in \mathbb{R}^{L \times (N + E)}$ should be encouraged to produce larger responses to $\tilde{v}$ to improve learning. We initialize the expansion of FC2 layer $W'_{exp}$ in the following manner:

$$W'_{InExp} = std\left(W'\right) \cdot u \cdot \Gamma$$

where $u \in \{-1, 1\}$ with probability 0.5 and $\Gamma$ is an amplification parameter. In our experiments we used $\Gamma = 2$. We found that this initialization technique is crucial in assuring convergence of the added weights and the ability of the new weights to improve classification results in low-shot setting.

