# OpenReview forum: "GENERATIVE LOW-SHOT NETWORK EXPANSION"
_ICLR.cc/2018/Conference — Reject_

### Official Review · AnonReviewer3 · 2017-11-26
**a paper proposing hard-distillation for few-shot learning**

**Rating:** 6
**Confidence:** 4

**Review:**

On few-shot learning problem, this paper presents a simple yet powerful distillation method where the base network is augmented with additional weights to classify the novel classes, while keeping the weights of the base network unchanged. Thus the so-called hard distillation is proposed. This paper is well-written and well organized. The good points are as follows,

1. The paper proposes a well-performance method for the important low-shot learning problem based on the transform learning.
2. The Gen-LSNE maintains a small memory footprint using a generative model for base examples and requires a few more parameters to avoid overfitting and take less time to train.
3. This paper builds up a benchmark for low-shot network expansion.

There are some problems,
1. There still is drop in accuracy on the base classes after adding new classes, and the accuracy may still drop as adding more classes due to the fixed parameters corresponding to the base classes. This is slightly undesired.
2. Grammatical mistake: page 3, line 5(“a additional layers”)

---

> ### Author Response · Authors · 2018-01-04
> **Regarding the Drop in accuracy on base classes**
>
> Thank you for emphasizing the topic of performance drop when adding new classes.
> The drop in the base classes is common in all of the competing methods, e.g. the  Soft-Dis/ICarl Adaptation where the base classes weights are adapted.
> It is expected that as the number of categories increase, and classes that are similar to one another are introduced, there will be a drop in classification accuracy, even for a fully trained network. Please notice that the proposed method outperforms the competing methods in most cases.

---

### Official Review · AnonReviewer2 · 2017-11-27
**Interesting, yet (by far) not sufficiently explored**

**Rating:** 4
**Confidence:** 3

**Review:**

The goal of this paper is to study generalisation to novel classes. This paper stipulates some interesting ideas, using an idea of expansion layers (using a form of hard distillation, where the weights of known classes are fixed), a GMM to model the already learned classes (to reduce storage), and a form of gradient dropout (updating just a subset of the weights using a dropout mask). All of these assume a fixed representation, trained on the base classifier, then only the final classification layer is adjusted for the novel examples.

The major drawback is that none of these ideas are fully explored. Given fixed representation, for example the influence of forgetting on base classes, the number of components used in the GMM, the influence of the low-shot, the dropout rate, etc etc.  The second major drawback is that the experimental setting seems very unrealistic: 5 base classes and 2 novel classes.

To conclude: the ideas in this paper are very interesting, but difficult to gather insights given the focus of the experiments.

Minor remarks
- Sect 4.1 "The randomly ... 5 novel classes" is not a correct sentence.
- The extended version of NCM (4.2.1), here uses as prototype-kNN (Hastie 2001) has also been explored in the paper of NCM, using k-means per class to extract prototypes.
- Given fixed representations, plenty of work has focused on few-shot (linear) learning, this work should be compared to these.

---

> ### Author Response · Authors · 2018-01-04
> **Experiments focus**
>
> We thank the reviewer. We would like to draw the reviewer’s attention to the following points:
>
> “Influence of forgetting on base classes:“
> The average accuracy of the base classes is presented in APPENDIX B BASE & NOVEL CLASSES TEST ERROR and in Figure 6,7,8,  In which we demonstrate the accuracy of the base classes as a function of the number of novel samples. We present the base class and novel class accuracy in each of the designed scenarios: generic classes from imagenet, Domain specific with similar novel classes,  Domain specific with similar class in base.
>
> “Influence of the number of components used in the GMM:”
> In section 4.5 RESULTS: DEEP-FEATURES GMM EVALUATION
> we’ve explored the effect of different number of GMM components on the ability to restore and re-create the accuracy on the base classes. We’ve explored the effect of GMM components on 5 imagenet based dataset and 3 based UT-Zappos50K dataset.
>
> “The influence of the low-shot:”
> In Sections 4.3 and 4.4 the effect of the number of novel samples on the performance is presented. We’ve experimented with a range of number of novel samples.
>
> “The dropout rate:”
> In Section 4.2.3 GRADIENT DROPOUT, we describe the experiment to evaluate the effect of Gradient Dropout in Low-Shot Expansion. We compare the results obtained with and without the use of gradient dropout, those are presented in Figures 2,3,4. Due to the already dense experimental section we did not add experiments done on various dropout ratios, though we agree that this is an interesting question.
>
>
> “The second major drawback is that the experimental setting seems very unrealistic: 5 base classes and 2 novel classes:”
> Please see answer above (to first reviewer) regarding the design of benchmark.

---

### Official Review · AnonReviewer1 · 2017-11-27
**The paper proposes a network/classifier expansion method to learn to classify with additional novel  classes in the future, without re-training with all the original data. It fine tunes the new parameters added with the new data (from novel classes), and with sampled examples from  simple generative models of the old classes. Overall the paper has a simple idea which is validated on limited settings (~10 classes only).**

**Rating:** 4
**Confidence:** 4

**Review:**

The paper proposes a method for adapting a pre-trained network, trained on a fixed number of
classes, to incorporate novel classes for doing classification, especially when the novel classes
only have a few training examples available. They propose to do a `hard' distillation, i.e. they
introduce new nodes and parameters to the network to add the new classes, but only fine-tune the new
networks without modifying the original parameters. This ensures that, in the new expanded and
fine-tuned network, the class confusions will only be between the old and new classes and not
between the old classes, thus avoiding catastrophic forgetting. In addition they use GMMs trained on
the old classes during the fine-tuning process, thus avoiding saving all the original training data.
They show experiments on public benchmarks with three different scenarios, i.e.  base and novel
classes from different domains, base and novel classes from the same domain and novel classes have
similarities among themselves, and base and novel classes from the same domain and each novel class
has similarities with at least one of the base class.

- The paper is generally well written and it is clear what is being done
- The idea is simple and novel; to the best of my knowledge it has not been tested before
- The method is compared with Nearest Class Means (NCM) and Prototype-kNN with soft distillation
  (iCARL; where all weights are fine-tuned). The proposed method performs better in low-shot
  settings and comparably when large number of training examples of the novel classes are available
- My main criticism will be the limited dataset size on which the method is validated. The ILSVRC12
  subset contains 5 base and 5 novel classes and the UT-Zappos50K subset also has 10 classes. The
  idea is simple and novel, which is good, but the validation is limited and far from any realistic
  use. Having only O(10) classes is not convincing, especially when the datasets used do have large
  number of classes. I agree that this will not allow or will takes some involved manual effort to
  curate subsets for the settings proposed, but it is necessary for being convincing.

---

> ### Author Response · Authors · 2018-01-04
> **Design considerations of the proposed benchmark**
>
> We would like to thank the reviewers for their commentary. We ask to draw the reviewer's attention to the following:
>
> In this work, we focus on a robotic unit in real-life scenarios. It is often desired to be able to adapt to a single or two new classes available with only very few samples. We choose to establish the proposed benchmark for the task of Low-Shot Network expansion in a manner that will reflect this task. That is, we wanted the number of novel classes to be relatively small so the effect of Network adaptation to small quantities of new data can be studied.
>
> We defined the class (base + novel) average accuracy to be our performance metric.
> We did not want the base classes accuracy to have overwhelmly more weight than the novel classes in the average accuracy metric, hence we defined the number of base classes to be in a range similar to the number of novel classes. We’ve composed a dataset with a common cardinality (similar to CIFAR10 is O(10), MNIST is O(10) and SVHN is O(10)). While O(10) is considerably smaller than imagenet 1000 classes classification task. It is still a common classification task, which is also common in robotic unit applications.
>
> In order to avoid bias in the constructed dataset, we’ve performed numerous random experiment, and reported their average result:
>
> The test on imagenet partitions was done by randomly selecting 50 classes, and then randomly partitioning to 5 groups of 10 , which are then further randomly partitioned to 5 base and 5 novel classes. The result of every experiment done with a given number of novel samples is averaged on 25 = 5X5 trails. That is Figure 2 is the result of 125 tests = 25 averaging X 5 #novel samples.
>
> Figure 5a Is the result of 25 (base,novel group avg) X 8 #novel samples = 200 trails.
>
> In this unbiased test case, we addressed 3 typical scenarios: generic classes from imagenet, Domain specific with similar novel classes,  Domain specific with similar class in base. We’ve further explored the effect and gain of Gen-LSNE compared to other methods in each of these scenarios. We concentrated our effort to analyze and explore the qualities of the proposed method in scenarios that are common in robotic unit real-life scenarios, and to the best of our knowledge the designed benchmark realistically reflect those.

---

### Decision · Program_Chairs · 2018-01-29
**ICLR 2018 Conference Acceptance Decision**

**Decision:**

Reject

**Comment:**

Two reviewers recommended rejection, and one is slightly more positive. The main concern is that the experiments are not convincing (ie, the number of base and added classes is very small). Furthermore, while the paper introduces several interesting ideas, the AC agrees with the second reviewer that each of these could be explored in more detail. This work seems preliminary. The authors are encouraged to resubmit to a future conference.